# Development of a discrete choice experiment questionnaire to elicit preferences by pregnant women and policymakers for the expansion of non-invasive prenatal screening

Hung Manh Nguyen[1], Carmen Lindsay[2‡], Mohammad Baradaran[3‡], Jason Robert Guertin[1,2‡], Leon Nshimyumukiza[4,5‡], Bounhome Soukkhaphone[1‡], Daniel Reinharz[1]*

1 Département de Médecine Sociale et Préventive, Université Laval, Québec, Canada, 2 Centre de Recherche du CHU de Québec, Université Laval, Québec, Canada, 3 Département de Génie Électrique et de Génie Informatique, Université Laval, Québec, Canada, 4 Institut national D'excellence en Santé et en Services Sociaux, Québec, Canada, 5 Faculté des Sciences Infirmières, Université Laval, Québec, Canada

☯ These authors contributed equally to this work.
‡ CL, MB, JRG, LN and BS also contributed equally to this work.
* Daniel.Reinharz@fmed.ulaval.ca

**Data Availability Statement:** All relevant data are within the paper and its Supporting Information files.

## Abstract

### Objective

An instrument for measuring intervention preferences applicable to both patients and policymakers would make it possible to better confront the needs of the supply and demand sides of the health care system. This study aimed to develop a discrete choice experiments (DCE) questionnaire to elicit the preferences of patients and policymakers. The instrument was specifically developed to estimate preferences for new conditions to be added to a screening program for fetal chromosomal anomalies.

### Methods

A DCE development study was conducted. The methods employed included a literature review, a qualitative study (based on individual semi-structured interviews, consultations, and a focus group discussion) with pregnant women and policymakers, and a pilot project with 33 pregnant women to validate the first version of the instrument and test the feasibility of its administration.

### Results

An initial list of 10 attributes was built based on a literature review and the qualitative research components of the study. Five attributes were built based on the responses provided by the participants from both groups. Eight attributes were consensually retained. A pilot project performed on 33 pregnant women led to a final instrument containing seven attributes: 'conditions to be screened', 'test performance', 'moment at gestational age to

**Funding:** Financial support for this study was provided entirely by a grant from the PEGASUS2 project funded by Génome Canada (with Génome Québec; Génome BC; Génome Alberta; the Québec Ministère de l'Enseignement supérieur, de la recherche, de la science et de la technologie; the Fonds de recherche Québec – Santé; la Fondation de l'Université Laval; and the Centre de recherche du CHU de Québec) – Grant number: LSARP2012-4523, and the Canadian Institutes for Health Research - Grant number: GPH129342. The funders had no role in study design, data collection and analysis, decision to publish, or preparation of the manuscript. Jason R Guertin is a FRQS Research Scholar Junior 1 (Award #266460).

**Competing interests:** The authors have declared that no competing interests exist.

**Abbreviations:** CHUL, CHU de Québec–Université Laval; DCE, Discrete Choice Experiments; HTSA, Health Technology Assessment; INESSS, Institut National d'Excellence en Santé et Services Sociaux; ISPOR, International Society for Pharmacoeconomics and Outcomes Research; NIPS, Non-Invasive Prenatal Screening.

obtain the test result', 'degree of test result certainty to the severity of the disability', 'test sufficiency', 'information provided from test result', and 'cost related to the test'.

## Conclusion

It is possible to reach a consensus on the construction of a DCE instrument intended to be administered to pregnant women and policymakers. However, complete validation of the consensual instrument is limited because there are too few voting members of health technology assessment agencies committees to statistically ascertain the relevance of the attributes and their levels.

## 1. Introduction

In a health technology assessment (HTA), economic evaluation is a methodological approach that makes it possible to judge on the economic value of a health intervention [1]. In most developed countries, the judgement should reflect the preferences of the patients/population for the intervention and/or its outcomes [2]. Two main approaches used in economic evaluation, cost-benefit and cost-utility analyses, have been developed to specifically produce scores that express these preferences.

These preferences may differ from those of policymakers at the political level, such as experts who are members of HTA agency committees and make final recommendations on technology adoption. However, these experts may value the technology differently compared to the patients/population, based on considerations that tend to speak of the broad societal values that underlie health care. Therefore, these preferences are not generally measured. One reason is the lack of measurement instruments of preferences applicable to both patients/population and policymakers.

In areas characterized by costly technological innovations and difficult ethical considerations, having the ability to measure the preferences of these groups (i.e., patients and policymakers) is even more relevant. This is the case for prenatal screening for fetal chromosomal abnormalities, a field that has benefited greatly from technological innovation with the advent of non-invasive prenatal screening (NIPS). NIPS tests have been introduced in many public programs, such as those in Belgium, the Netherlands, France, and Denmark. In Canada, where the responsibility of offering health services rests on provinces, several health jurisdictions have made NIPS accessible in publicly funded programs for high-risk pregnancies [3]. Clinically, NIPS allows the measurement of cell-free fetal DNA in the maternal blood to detect a large array of chromosomal anomalies [4]. However, the list of anomalies currently detected by NIPS in public sector tend to be limited to trisomy 21 (Down syndrome), trisomy 13 (Patau syndrome), and trisomy 18 (Edward's syndrome).

The discrete choice experiment (DCE) is a methodological approach increasingly used to elicit preferences for health and healthcare interventions [5]. The approach, which is theoretically rooted in the random utility theory, allows for the exploration of the relative importance of different attributes and levels within a decision-making process [6]. In the field of prenatal screening, DCE allows measuring preferences for perinatal interventions that concern a fetus [7] or newborn from the perspective of adults, solving the problem of the impossibility of defining utility for a fetus [8]. Moreover, a cost attribute related to the test can be included among the DCE attributes, thereby quantifying preferences in term of monetary value [9, 10]. Another advantage of this method is that a common DCE instrument could be used to

measure the preferences for perinatal intervention from the perspective of various stakeholders, such as populations/patients [11–14] and other groups, such as health professionals [15–18], policymakers, or those who work in HTA agencies. Information provided by such a quantitative instrument, which can be administered to a representative sample of stakeholders, is expected to provide evidence of alignments and disagreements between groups of actors, which would complement information coming from qualitative studies.

In the field of prenatal screening, DCE studies have revealed different preference levels among pregnant women, their partners, and health care professionals [15, 16, 18–21]. However, little is known about how the preferences of pregnant women differ from those of policymakers. In this paper, we report the development of a DCE instrument to elicit the preference levels of pregnant women and policymakers, for the expansion of a NIPS-based public prenatal screening program for the detection of fetal chromosomal anomalies.

## 2. Methods

A growing consensus in the literature and empirical evidence suggest that credible attributes and attribute levels for a DCE must be policy-relevant, important to the study population, and consistent with the random utility theoretical foundation of the DCE [6, 22]. In line with these requirements, the development of attributes and attribute levels for this DCE questionnaire was undertaken in sequential steps based on best practice guidance [6, 23–25]. This process began with a systematic review of the literature. The review was followed by a qualitative study undertaken with pregnant women and policymakers to identify and select attributes and attribute levels for the DCE instrument. A pilot study was conducted to test the DCE instrument and further validate the list of attributes and attribute levels.

Ethical approval for this study was obtained from the teaching hospital ethics committee (*Comité d'éthique de la recherche du CHU de Québec-Université Laval*, project 2020–4877), and permission was granted to enroll participants at the CHUL (*Centre Hospitalier de l'Université Laval*) in Québec City, Canada. Written informed consent was obtained from all participants. Pregnant women signed consent form after being informed of the study by the two researchers (HMN and CL). Policymakers provided their electronic signatures on the consent form, which was returned to the researchers via email.

### Attributes and levels identification

The attribute identification process began with a literature review, followed by in-depth interviews with pregnant women and policymakers.

A systematic review of the literature on the use of DCE in the field of prenatal screening for chromosomal anomalies was conducted by two researchers (HMN and BS), according to the Preferred Reporting Items for Systematic Reviews and Meta-Analyses Protocols 2015 (PRISMA-P 2015) recommendations [26, 27]. A librarian validated the search strategy. Six databases (PubMed, Embase, EconLit, Cochrane Library, Web of Science, and ProQuest Dissertations and Theses Global) were searched from their inception to February 2019. MESH terms and text words were used to target studies using DCE for prenatal tests for fetal conditions, including: 'discrete choice experiment', 'discrete choice experiments', 'discrete choice model', 'discrete choice models', 'discrete choice modelings', 'discrete choice conjoint experiment', 'pathworth utilities', 'functional measurement', 'paired comparison', 'pairwise choices', 'conjoint analysis', 'conjoint measurement', 'conjoint study', 'conjoint studies', 'conjoint choice experiment', 'conjoint choice experiments' and 'stated preference'. As suggested by Soekhai *et al.*, we excluded the search terms "DCE" and "conjoint" since they were reported to yield too many irrelevant results. Attribute terms and their valuation were also used to target DCE conducted

in prenatal care [28]. The reference lists of the retrieved studies were also scanned for any additional relevant publications that were not found in the search. The search was expanded to include empirical DCE conducted with policymakers.

The identified publications were screened according to the following inclusion criteria: having a target population consisting of policymakers, pregnant women, or women of child-bearing age; having received a DCE questionnaire on prenatal tests for fetal conditions; and having a focus on DCE attributes. The articles or other documents had to be complete and be in English. There was no restriction of geographical areas applied. Any disagreement regarding the selection of an article was discussed until a consensus was reached. A third researcher (DR), if necessary, was implicated.

The International Society for Pharmacoeconomics and Outcomes Research (ISPOR) checklist items [23] were used to assess the methodological quality of the retained documents and extract data. As the review of literature aimed primarily at identifying the attributes considered in DCE studies, we considered the 9 first ISPOR items (research questions, attributes and levels, construction of tasks, experimental design, preference elicitation, instrument design, data collection, statistical analysis, results, and conclusions) and not the last one (study presentation).

Data for each retained study were independently extracted and summarized by the two reviewers (MHN and BS). Extraction was about the ISPOR checklist items, to which, one additional item was added: main findings. Any disagreement was solved by discussion and debate. The process of consensus search involved the third researcher (DR).

This systematic review of the literature led to the identification of attributes and levels for prenatal screening preferences (a fully unpublished report is presented in S1 Text). This review identified potential attributes that have been found to influence preferences for undertaking a screening test for a new condition in a NIPS-based prenatal screening program.

A qualitative study that conceptually lay on the review of the literature was then undertaken to test the attributes suggested from the literature and identify others that would be important to both groups (i.e., pregnant women and policymakers) regarding the addition of new conditions to a public prenatal screening program for fetal chromosomal anomalies.

Semi-structured interviews based on an interview guide (see S2 Text), were conducted with pregnant women (*n = 12*) and policymakers (*n = 4*) between February and August 2020. This guide was pre-tested with two pregnant women and one policymaker. These three individuals were not included in the study sample. They were asked to comment only on their understanding of the interview guide.

The inclusion criteria for pregnant women were primigravida, aged 18 years or older, and consulting the obstetric department of the CHUL in Québec City, Canada, for their first prenatal echography. The CHUL is one of two public hospitals in Québec City where obstetrical follow-up is provided to the population regardless of any pregnancy-associated risks. The inclusion criteria for policymakers were members of a permanent scientific committees of Québec's HTA Agency, INESSS (*Institut National d'Excellence en Santé et Services Sociaux*), or an executive at the Ministry of Health and Social Service (Québec); and involvement in decision-making on maternal-infant health services in the province. Eligible participant consent was obtained before interviews were conducted.

Interviews were structured in such a way that, at the beginning, respondents were encouraged to say whatever they thought was an important characteristic to consider when deciding which condition to screen for, without being interrupted or influenced. Depending on the spontaneously produced information, specific questions in the interview guide were asked regarding dimensions that were not discussed. For example, participants were asked how much they would be willing to pay out-of-pocket to undergo test screening for additional

conditions rather than the three common ones (trisomy 21, 18, and 13). This question reflects the current situation in Canada, where conditions other than those listed in the prenatal screening program can be screened but at a cost that must be paid out-of-pocket. All interviews were digitally recorded. Verbatim responses were transcribed using NVivo Transcription (QRS International 2020, Burlington, Massachusetts). The transcriptions were independently checked by two researchers (HMN and CL) while relistening to the recordings.

The analysis was performed independently by researchers HMN and CL. It aimed to identify key attributes and their levels. The initial framework (a pre-established coding scheme) used to guide the identification of the attributes was based on the interview guide (deductive approach) [29, 30]. Additional codes were generated where required (inductive approach) [29, 30].

The analysis was first based on triangulation (i.e., categories and themes derived from several sources of information) [31]. The two researchers coded all interviews independently. In case of disagreement, discussions were held until a consensus was reached on the final coding.

The attributes identified by the two respondent groups were merged. All possible attribute levels arising from the interview were retained. The attributes defined by only one group were also included in the list of attributes/levels for the subsequent selection procedure.

## Attribute/Level selection and framing

Attribute selection followed a Delphi process aimed at finding a consensus between both groups of participants on the set of attributes and levels to be included in the DCE instrument [32, 33]. The Delphi approach involved consultations and a focus group discussion. After each step, discussions were held between research team members to refine the list of attributes and levels.

First, the consultation process was undertaken with the same participants who participated in the attribute/level identification step and agreed to be contacted for future study. A list of potential attributes and levels retrieved from the previous codification procedure was presented to the participants. This consultation process aimed to refine the first list of potential attributes and levels. This list was sent to each participant via e-mail to explore their opinions regarding the meaning and relevance of the attributes and levels. Participants were also asked to provide justification if they considered any attribute to be irrelevant. Attributes considered relevant by the participants were retained and modified if required, whereas those considered irrelevant were excluded.

In the second step, the list of retained attributes was refined based on a focus group discussion conducted with three pregnant women and one policymaker solicited from the same hospital (A focus group discussion with policymakers alone was not held because of the limited pool of potential participants). The focus group was held at the CHUL hospital. The four participants were asked to provide their opinions on the relevance of the attributes and levels.

Information gained from the selection process was synthesized by one researcher (HMN). The content of the attributes and levels was then revised and framed by both research team members to ensure their relevance and comprehension of the wording.

## Pilot testing the DCE survey

A pilot project was undertaken to explore the feasibility of the survey to be administered to a large sample of respondents (i.e., to explore the understanding of the tasks, the complexity of the choices, and the time needed to fill the questionnaire), to generate prior parameters for the DCE design, and to assess the statistical relevance of the attributes and levels of the first version of the final questionnaire.

The pilot project confirmed the attributes to be added to the full-scale study where the plausibility of choices will be tested with pregnant women and policymakers.

**Construction of choice tasks.** The DCE design and construction of tasks followed the 10-points ISPOR checklist for best practice guidance for conjoint experiment design [23]. An experimental design was performed to generate unforced choice questionnaires [34]. The instrument consists of a series of choices. Each choice has two options that differ in their levels for each attribute. These attributes represent the characteristics of a hypothetical additional test that can be added to the list of tests already included in public prenatal screening programs.

Generic labels were used to identify the options, called tests A and B, across all choice sets. Respondents must choose their preferred option or declare that they cannot decide which of the two options is preferred.

Moreover, a D-optimal design [35, 36] was used to create a pairwise DCE pilot project (two x two scenarios per choice task). Details of the sample calculation for this pilot project can be found in S3 Text.

**Sampling and recruitment.** For practical reasons (limited pool of potential respondents in the group of policymakers), we could only test the questionnaires in this pilot project with pregnant women (between July and September 2021).

Women enrolled in our study were recruited from a clinical trial on NIPS (PEGASUS-2, ClinicalTrials.gov Identifier: NCT03831256) and agreed to participate in an additional study. The 28th-30th weeks gestational age was chosen to avoid interference with the clinical trial data collection periods.

At this stage, participants were asked to choose the screening option that either suited them the best in each task or state that they could not express a preference.

**Data collection and analysis.** Choices in the DCE choice tasks were automatically collected by using Université Laval's (Canada) LimeSurvey platform. Eligible participants received an invitation containing information regarding the nature of the study and a link that led them to participate. Once the informed consent form was given by clicking on the "*accept*" function, participants had access to the DCE pilot survey.

Participants were given two weeks to complete the survey. After two weeks, the link to the unanswered questionnaire was deactivated and sent to newly solicited participants until the sample size had been reached.

A DCE study assumes that individuals select the alternative that provides the highest utility (random utility maximization theory) [37]. The preference data were codified in an Excel spreadsheet for analysis using a conditional logit model (SAS, release 9.4; Cary, North Carolina) to estimate the relative importance (utility) of each attribute on the preferences of the participants. The model is expressed as follows:

$$V = ß0 + ß1\,condition\_1 + ß2\,test\_performance\_2 + ß3\,result\_moment\_1 + ß4\,test\_certainty\_2 \\ + ß5\,test\_sufficiency\_2 + ß6\,info\_provided\_to\_women\_1 + ß7\,info\_provided\_to\_women\_2 \\ + ß8\,geographic\_2 + ß9\,geographic\_3 + ß10\,cost + \varepsilon$$

where $V$ is the utility derived from a given hypothetical screening test (choice) and $\varepsilon$ refers to the error term. All attributes were considered as factors (dummy coded) except for the cost related to the test attribute, which was treated as continuous. Coefficients $ß_1$ to $ß_{10}$ generated by the regression analysis were given a "+" or "-" to indicate the direction of the preference for each attribute. The significance and size of the coefficients were used to estimate the relative importance of the attributes on choice. The significance level was fixed at 0.25% [38].

## 3. Results

### Attributes/Levels identification

**Review of the literature.**   Of the 58 DCE publications found during the database search for the systematic review of the literature, 13 met our eligibility criteria (see S1 Text for full report). All studies were conducted in high-income countries. While these countries provided their populations with a fully/partially universal health coverage, their jurisdictions varied in terms of patients' financial contribution to prenatal care. Moreover, none of these studies had been conducted with policymakers, i.e., individuals who voted for the offer of prenatal care intervention.

Some attributes appeared quite frequently, notably detection rate (11 studies), risk of miscarriage (nine studies), information content (seven studies) and time in pregnancy when results received (six studies). A cost attribute was included in five DCE studies.

Differences in preferences for different levels of the attributes between woman and other stakeholders (i.e., partners, obstetricians, gynecologists, genetic counselors, clinical geneticists, nurses, and midwife) were presented. Women tended to highly value risk of miscarriage and reception of comprehensive information, while health professionals tended to place higher value on detection rate of the test, moment during pregnancy when test results would become available and capacity to diagnose a condition earlier on. Furthermore, in studies whose DCE questionnaires included a cost attribute, trade-offs were reported between screening options presented to respondents in terms of willingness-to-pay values.

**Qualitative study.**   Table 1 details the identification process of attributes and levels through a qualitative study. The process resulted in the identification of nine categories that could potentially become attributes of a test that should influence, according to the respondents, the decision to expand the list of conditions screened for in a NIPS-based prenatal screening program. Each category and its sources contained multiple decisional points suggesting potential levels for each attribute. This step showed that pregnant women and policymakers have both common and unique preoccupations.

Indeed, several preoccupations were shared by both groups, notably regarding the need to select conditions in the list of chromosomal anomalies that can be screened by NIPS, the type of information resulting from the test that should be transmitted to parents, the test performance, the parents' out-of-pocket costs related to the test, and the level of certainty of the test result. Preoccupations regarding these categories may differ between pregnant women and policymakers. For example, pregnant women desired to avoid excluding conditions, while policymakers were concerned with limiting the choice of conditions in which disability can be prevented or mitigated by medical intervention. Sensitivity and specificity (false negative rate and false positive rate) issues were raised by all four policymakers but not by pregnant women. Policymakers, unlike women, did not raise the question of gestational age to receive the results of the test or the question of 'geographical access to the test'. They tended to place greater emphasis on the complexity of the test procedure and discussed the issue of stress that the test might cause for women.

Table 2 displays the initial list of ten attributes based on the identification of categories and decisional points. Five attributes ('conditions to be screened', 'information provided from test result', 'test uncertainty', 'cost related to the test', and 'test performance') were built from what was said by both groups of participants. Two attributes were built from what was said by women only ('moment at gestational age to obtain the test result' and 'geographical access to the test') and three from what was said by policymakers only ('test sensitivity', 'test specificity' and 'test procedure'). Based on the responses provided by a single group, these attributes were retained by the researchers on the basis that they could be of interest to another group. This hypothesis was tested in the next step of the instrument construction.

**Table 1. Attributes/levels identification.**

| Category | Decisional points (sources) | Attributes | Levels | | Potential consensus | |
|---|---|---|---|---|---|---|
| | | | Pregnant women | Policymakers | Attribute | Levels |
| **1. Conditions being screened** | • Conditions that women are interested to know (PM)<br>• Detect as many as possible conditions (PW)<br>• Detecting severe conditions/anomalies (PW, PM)<br>• The diseases can be confirmed (PM)<br>• Conditions that could also be screened at newborns (PM) | **Conditions to be screened** | 1. Detect as many as possible conditions<br>2. High prevalent genetic conditions (not rare) | 1. Conditions that women are interested to know<br>2. Severe conditions that being able to be confirmed by other tests. Treatments or interventions are available | **Conditions to be screened**\* | 1. As many as possible<br>2. Non-rare diseases<br>3. Lead to treatment or interruption |
| **2. Information provided from test result** | • Reliable and comprehensive information (PW, PM)<br>• Easy to interpret the results (PM)<br>• Simple information (PW) | **Information provided from test result** | 1. As much as possible on this risk/probability that baby might have genetic problems<br>2. Simple information | 1. Reliable and comprehensive information; Results are easily interpreted<br>2. Simple information. Difficult to make prognosis | **Information provided from test result**\* | 1. Information on risk of disability<br>2. Medical implications<br>3. Social implications |
| **3. Test performance** | • Low false positive (less than 5%) and false negative rates (PM, PW)<br>• Accept higher false positive rate (PM)<br>• Take the test regardless detection rate to reduce risk of any possible problems to the baby (PW) | **Test performance** | 1. Not consider a test with false positive rate higher than 5%<br>2. Detection rate: 50%-95%<br>3. Detection rate: above 95%<br>4. Take the test regardless detection rate to reduce risk of any possible problems to the baby | 1. Low false positive and false negative rates<br>2. Higher false positive rate to maximize number of cases | **Test performance**\*\* | 1. known<br>2. uncertain |
| | | | | | **Test sensitivity**\*\*\* | 1. Equal or less than 75%<br>2. 76%-85%<br>3. 86%-95%<br>4. Equal or above 96% |
| | | | | | **Test specificity**\*\*\* | 1. Equal or less than 75%<br>2. 76%-85%<br>3. 86%-95%<br>4. Equal or above 96% |
| **4. Test uncertainty implication** | • Women stress/anxiety and public system cost (PM)<br>• Difficult to make confirmation and prognosis (PM) | **Test uncertainty** | 1. Not consider a test with low accuracy rate of severity<br>2. Take the test regardless the uncertainty level (neither physical nor intellectual problems)<br>3. Prefer a test with higher accuracy rate of intellectual problems | 1. Not likely to accept low level of certainty regardless health outcomes: physical or intellectual<br>2. Accept uncertainty but depending on severity of screened conditions | **Test uncertainty**\* | 1. Regardless the uncertainty<br>2. Accept low accuracy rate on physical problems<br>3. Not taking test with the uncertainties |
| **5. Acceptance of uncertainty level related to physical and intellectual problems** | • Not accept low level of certainty (PW, PM)<br>• Depend on individuals (PM) | | | | | |
| **6. Test procedure** | • Simple and rapid (PM)<br>• Women' stress (PM) | **Test procedure** | | 1. Test procedure and confirmation tests are simple and rapid which are not resulting in stresses on women<br>2. Complex test procedure causing stress on women | **Simplification of test procedure**\*\*\* | 1. simple<br>2. complex |
| **7. Cost related to the test** | • Reasonable cost accepted by women (PM)<br>• At a reasonable cost (PW)<br>• Publicly funded tests (PM) | **Cost related to the test** | 1. C\$ 0 (Cost-free access)<br>2. C\$ 100—C\$ 500<br>3. Willing to pay to have information of test regardless the amount of money | 1. Affordable cost accepted by pregnant women<br>2. Reasonable cost for public system, not overcome threshold | **Cost related to the test**\* | 1. C\$ 0<br>2. C\$ 100<br>3. C\$ 200<br>4. C\$ 300 |

(*Continued*)

**Table 1.** (Continued)

| Category | Decisional points (sources) | Attributes | Levels | | Potential consensus | |
|---|---|---|---|---|---|---|
| | | | Pregnant women | Policymakers | Attribute | Levels |
| **8. Moment in gestational age to obtain test result** | • As soon as possible (PW)<br>• 8–10 weeks (PW)<br>• 12 weeks (PW) | **Time to result** | 1. As soon as possible<br>2. 8–10 weeks<br>3. After 10 weeks | | **Time to result**\*\* | 1. Before the third regular pregnancy visit<br>2. Before the second (10–12 week) regular pregnancy visit |
| **9. Geographical assess to the test** | • Test can be accessible in the proximity, either hospitals or clinics (PW) | **Geographical access to the test** | 1. In a local health care facility<br>2. The confirmatory test is done in a regional hospital<br>3. The confirmatory test can only be done in a hospital located in a city center | | **Geographical access to the test**\*\* | 1. In a local health care facility<br>2. The confirmatory test is done in a regional hospital<br>3. The confirmatory test can only be done in a hospital located in a city center |

PW, pregnant women; PM, policymakers; *consensual attributes by pregnant women and policymakers

\*\*attributes by pregnant women

\*\*\* attributes by policymakers

## Attribute/level selection and framing

Attribute/level selection and framing were based on the modified Delphi process, including consultations (two policymakers and a pregnant woman) and a focus group discussion (one policymaker who participated in the in-depth interview and agreed to participate in future studies, and three pregnant women who had not participated in the first interviews). The participants suggested only one major change. They agreed that 'test specificity' and 'test sensitivity' concepts were difficult to understand by pregnant women and expressed in test performance attribute. The test specificity and test sensitivity attributes were therefore redundant.

The consultations and focus group discussion also led to the rewording of some attributes and levels to facilitate the comprehension of the questionnaire (for example, 'test procedure' became 'test sufficiency'). This process reduced the questionnaire to eight attributes. Five attributes had two levels and three attributes had three levels. Amongst the attributes, the levels of 'cost related to the test' was built based on what women said about how much out-of-pocket money they were prepared to give for the test. The details are presented in Table 3.

## DCE pilot results and final attributes/levels selection

A pilot DCE was constructed using the eight identified attributes. Of 115 invitations sent, 68 was initiated. The DCE pilot study was completed by 33 pregnant women (aged from 19–37 years), whereas those did not complete the survey mostly stopped at the informed consent form (78%). Each respondent was asked to complete 7 comparison choice tasks.

The pilot project revealed that participants took an average of 11 min to complete the questionnaire (minimum, 2 min; maximum, 49 min). Fig 1 illustrates an example of a choice task in the DCE pilot study (an example of the full survey on LimeSurvey is available upon request).

Table 4 shows that the 'test performance' attribute was the most contributive to the choices made, followed by 'test uncertainty' 'moment at gestational age to obtain the test results', 'test

**Table 2. List of potential attributes and levels for consultations and focused group discussion.**

| Attributes | Levels |
|---|---|
| 1. Conditions to be screened | 1. Can be treated or lead to a termination of pregnancy |
| | 2. Can be treated or lead to termination of pregnancy and should not be rare |
| 2. Information provided from test result (i.e., the test result that is presented to a pregnant woman is about) | 1. The risk of disability |
| | 2. The risk of disability and its medical implications |
| | 3. The risk of disability, its medical and social implications |
| 3. Test performance | 1. Known |
| | 2. Uncertain |
| 4. Test sensitivity | 1. Less than or equal 75% |
| | 2. 76 to 85% |
| | 3. 86 to 95% |
| | 4. Equal or above 96% |
| 5. Test specificity | 1. Less than or equal 75% |
| | 2. 76 to 85% |
| | 3. 86 to 95% |
| | 4. Equal or above 95% |
| 6. Moment at gestational age to obtain the test result | 1. Before the third prenatal visit |
| | 2. Before the second prenatal visit |
| 7. Cost related to the test | 1. C$ 0 |
| | 2. C$ 100 |
| | 3. C$ 200 |
| | 4. C$ 300 |
| 8. Test uncertainty (i.e., degree of test result certainty to the severity of the disability) | 1. The child will have a physical and/or intellectual disability |
| | 2. The child could have the disease, but without it resulting in a physical and/or intellectual disability |
| 9. Test procedure | 1. Simple |
| | 2. Complex |
| 10. Geographical access to the test | 1. In a local health care facility (e.g., CLSC) |
| | 2. The confirmatory test is done in a regional hospital |
| | 3. The confirmatory test can only be done in a hospital located in a city center |

sufficiency', 'cost related to the test', and 'information provided from test results' (p-values < 0.05). Two other attributes, 'conditions to be screened' and 'geographical access to the test', were the least influential to the choice made (p-values at 0.2218 and 0.3247 respectively). Furthermore, the analysis also revealed which attribute levels were preferred by pregnant women. These results allowed us to form a choice task containing a dominant option for the full-scale DCE study that will be used to evaluate if answers given are random or well-thought.

The conditional logit model found most attributes to be statistically significantly associated with the choice (p value < 0.05), except 'geographical access to the test' and 'conditions to be screened' (p values at 0.3247 and 0.2218 respectively). However, because the predefined threshold for retaining an attribute was fixed at 25% [38], only 'geographical access to the test' was excluded.

Statistical analyses did not suggest modifications to the attribute levels. However, researchers decided to extend the maximum cost paid by users of the test to C$ 1000 (the maximum

**Table 3. List of attribute and attribute levels for the pilot project.**

| Attributes | Levels | Explication |
|---|---|---|
| • Conditions to be screened | 1. Can be treated or lead to a termination of pregnancy | A test can detect as many conditions as possible, provided that in case of a positive result, medical intervention is then possible. |
|  | 2. Can be treated or lead to termination of pregnancy and should not be rare | A rare disease is defined as a condition that affects less than one in 200,000 individuals. This test would therefore make it possible to detect diseases that are rarer than Down syndrome, which affects 300 children out of 200,000 births. |
| • Test performance (degree of accuracy of the test result) | 1. Known | In a few cases, the result of a screening test is incorrect. When the percentage of the error is known, the mother can be told what the probability is, that a second test, called a confirmatory test, which is rarely wrong, will confirms or reject the first test result. |
|  | 2. Uncertain | In a few cases, the result of a screening test is incorrect. When the percentage of error is uncertain, the probability that a second, confirmatory test, which is rarely wrong, will confirm or invalidate the first test result cannot be specified. An uncertain result is common for rare diseases |
| • Moment at gestational age to obtain the test result | 1. Before or at the third prenatal visit | The result is communicated at the latest during the third prenatal visit, around the 24th week of pregnancy |
|  | 2. Before or at the second prenatal visit | The result is communicated at the latest, during the second prenatal visit, around the 18th week |
| • Degree of test result certainty to the severity of the disability | 1. The child is certain to have a severe physical and/or intellectual disability that will affect the child's quality of life | The result may detect a physical or intellectual problem that will lead to a severe disability that will affect the child's quality of life |
|  | 2. The child may have the disease. However, having the disease does not necessarily mean that the child will have a severe physical and/or intellectual disability | The result can detect an intellectual or physical problem but does not indicate the severity of the disability. |
| • Test sufficiency | 1. A positive result can be confirmed during regular prenatal visits | Screening interventions are offered to all women during a regular pregnancy visit |
|  | 2. A positive result may require confirmation by tests that are not offered during regular visits | Screening interventions may require additional interventions, such as additional visits or specific tests like amniocentesis |
| 2. Information provided from test result (i.e., the test result that is presented to a pregnant woman is about) | 1. The risk of disability | The information is about the possibility that the child may have a disability |
|  | 2. The risk of disability and its medical implications | The information is about the possibility that the child may have a disability and the medical consequences of the disability which may require treatment. |
|  | 3. The risk of disability, its medical and social implications | The information is about the possibility that the child may have a disability, the medical consequences of the disability, and the social impact of the disability on the life of the child and family. |
| 9. Geographical access to the test | 1. All screening services are provided at a local health facility (e.g., CLSC) | Screening can be done at a facility near the pregnant woman's residence |
|  | 2. If the test is positive, confirmatory testing is done at a regional hospital | Full testing (screening and confirmatory testing) may require a pregnant woman to go to the local referral hospital |
|  | 3. If the test is positive, confirmatory testing can only be done in a hospital located in a big city | Full testing (screening and confirmatory testing) may require a pregnant woman to go to a hospital in an urban center |
| 7. Cost related to the test | 1. C$ 0 |  |
|  | 2. C$ 150 |  |
|  | 3. C$ 300 |  |

paid in Canada for the detection of chromosomal anomalies in the private sector, year 2021) with an arbitrary C$ 200 increment (six levels) to reflect the real amount that women might have to pay for detecting anomalies not presently listed in a prenatal screening program, with NIPS.

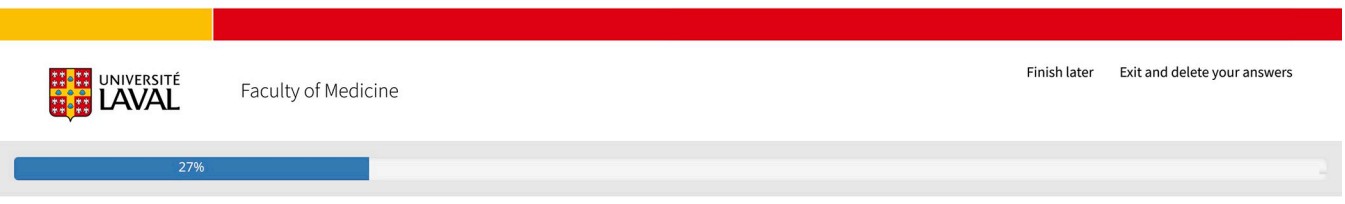

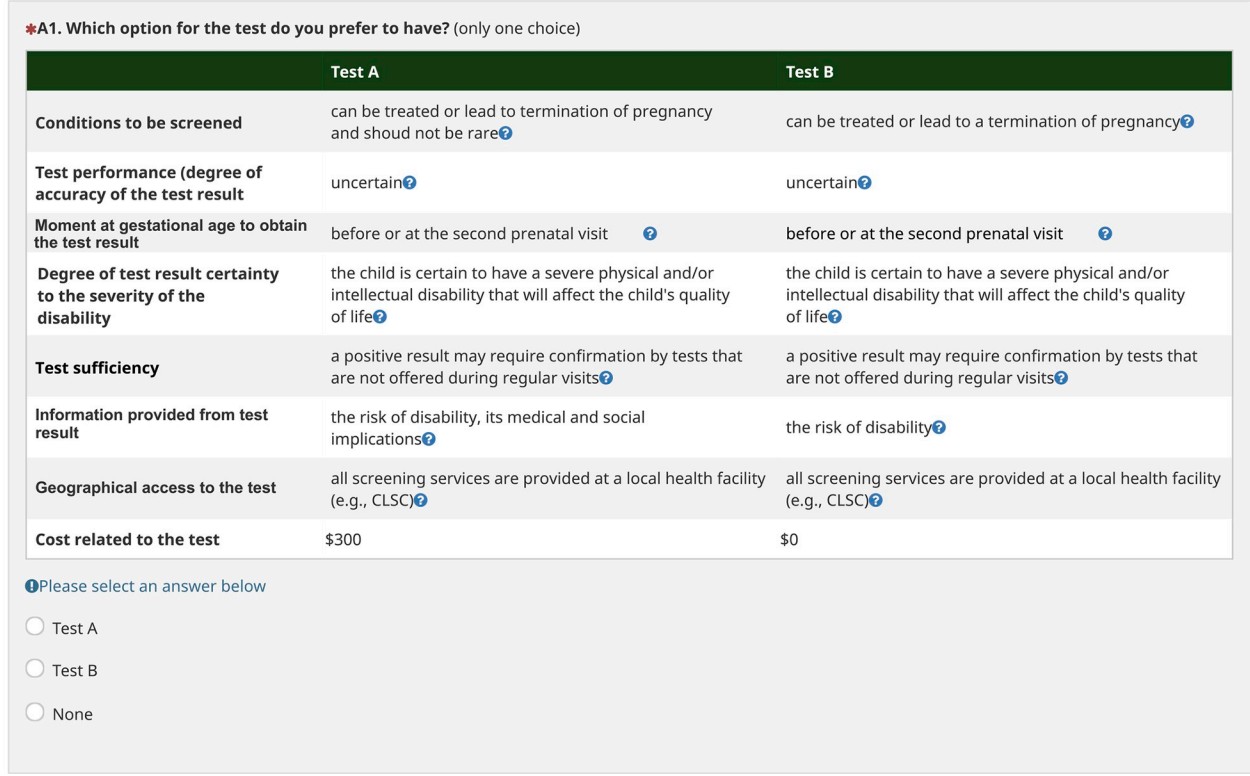

**Fig 1. Sample of a choice task in the DCE pilot testing project.**

The final DCE questionnaire contained seven attributes as detailed in Table 5 (five attributes with two levels, one attribute with three levels, and one attribute with six levels).

## 4. Discussion

The work presented in this paper concerns the construction of a DCE instrument that addresses the issue of adding a test to a prenatal screening program that can be administered to both pregnant women and policymakers. Having an instrument built on the preoccupations of both groups allows researchers to compare the preference levels for a new test from both sides. It also allows the decision-making process to include comparable information from

**Table 4. Conditional analysis of effects.**

| Attributes | Levels | Coefficient | (95% Confident Interval) | Pr > ChiSq |
|---|---|---|---|---|
| 1. Conditions to be screened | 1 | -0.3354 | (-0.8730 to 0.2026) | 0.2218 |
| | 2 | 0 | | |
| 2. Test performance (degree of accuracy of the test result) | 1 | 0 | | |
| | 2 | **-1.3176** | (-1.8617 to -0.7735) | **< .0001** |
| 3. Moment at gestational age to obtain the test result | 1 | **-0.7890** | (-1.3131 to -0.2649) | **0.0032** |
| | 2 | 0 | | |
| 4. Degree of test result certainty to the severity of the disability | 1 | 0 | | |
| | 2 | **-0.8860** | (-1.4230 to -0.3490) | **0.0012** |
| 5. Test sufficiency | 1 | 0 | | |
| | 2 | **-0.6799** | (-1.1867 to -0.1731) | **0.0086** |
| 6. Information provided from test result | 1 | **-1.0263** | (-1.7172 to -0.3354) | **0.0036** |
| | 2 | -0.3915 | (-1.0530 to 0.2700) | 0.2460 |
| | 3 | 0 | | |
| 7. Geographical access to the test | 1 | 0 | | . |
| | 2 | -0.0856 | (-0.7124 to 0.5412) | 0.7891 |
| | 3 | 0.3854 | (-0.2596 to 1.0304) | 0.2451 |
| 8. Cost related to the test | 1 | 0 | | |
| | 2 | -0.4190 | (-1.0913 to 0.2533) | 0.2218 |
| | 3 | **-1.1759** | (-1.8466 to -0.5052) | **0.0006** |

**Table 5. Final list of attributes and levels for full-scale DCE.**

| Attributes | Levels |
|---|---|
| 1. Conditions to be screened | 1. Can be treated or lead to a termination of pregnancy |
| | 2. Can be treated or lead to termination of pregnancy and should not be rare |
| 2. Test performance (degree of accuracy of the test result) | 1. Known |
| | 2. Uncertain |
| 3. Moment at gestational age to obtain the test result | 1. Before or at the third prenatal visit |
| | 2. Before or at the second prenatal visit |
| 4. Degree of test result certainty to the severity of the disability | 1. The child is certain to have a severe physical and/or intellectual disability that will affect the child's quality of life |
| | 2. The child may have the disease. However, having the disease does not necessarily mean that the child will have a severe physical and/or intellectual disability |
| 5. Test sufficiency | 1. A positive result can be confirmed during regular prenatal visits |
| | 2. A positive result may require confirmation by tests that are not offered during regular visits |
| 6. Information provided from test result | 1. The risk of disability |
| | 2. The risk of disability and its medical implications |
| | 3. The risk of disability, its medical and social implications |
| 7. Cost related to the test | 1. C$ 0 |
| | 2. C$ 200 |
| | 3. C$ 400 |
| | 4. C$ 600 |
| | 5. C$ 800 |
| | 6. C$ 1000 |

groups with different preoccupations but equally relevant concerns in the data used to support eventual decision.

Our instrument shows similarities in its attributes with other DCE instruments described in the literature and built for use in studies on prenatal screening of the fetus [14, 15, 17, 20]. All instruments found in the literature have attributes related to the level of information provided by the test results, the time in gestational age to receive the results of a screening test, and test sufficiency, that is, the impact of the test on the need for further invasive/non-invasive procedures to confirm a screening result. All instruments also have a 'test performance' attribute, presented sometimes in different terms as 'detection rate' or 'accuracy rate' [14, 15, 17, 20]. Unsurprisingly, this concern was also shared by policymakers involved in the construction of our instrument but described in terms of 'false-negative' and 'false-positive' rates.

However, two dimensions present in our DCE instrument are not found in other DCE instruments: one regarding which conditions should be screened and another regarding which certainty level of the test result regarding the severity of a disability could be accepted [13, 15, 16, 18, 39, 40]. The reason for the discrepancy is probably due to the fact that other DCE instruments focus on the use of tests to screen for common chromosomal anomalies, for which the performance of the test is high. Our instrument targets potential chromosomal conditions that are not currently screened but can be added to a screening program. The performance of the detection test for these conditions tends to be lower [41]. These conditions also tend to lead to a wide range of phenotypes and, hence, the severity of disability.

Moreover, the construction of the DCE instrument shows that a consensus has been reached on a final version, even if some attributes were initially proposed by only one group (test sensitivity and test specificity). This probably reflects a changing understanding of some attributes that were of little concern before the study, but whose perceived relevance grew over the course of the study. To our knowledge, only one published DCE instrument in the field of pharmacy subsidy decisions has been administered to both patients and policymakers [42]. However, this study has some uncertainties regarding whether both public and health policymakers/experts were effectively involved in the identification of the questionnaire attributes. Furthermore, this study does not provide information regarding whether or how consensus on the DCE instrument attributes between the two groups has been examined.

This study has some limitations. First, about half the individuals who agreed to participate (51%) did not complete the survey. Of these, 78% quit the survey after reading the informed consent form, whereas the rest (22%) stopped answering the first question. The reasons why many potential participants refused to take part in the study were not investigated. However, this observation is commonly reported in DCE studies. The main reasons for the low participation rate presented in the literature are the complexity of the task asked, sensitivity of the study topic, cognitive burden of the DCE questionnaire, and having poor familiarity with trade-off tasks [43, 44].

Second, we were unable to test the instrument with policymakers in the pilot project because of the limited pool of individuals who voted as members of health policy deliberating committees. Indeed, the recruitment of policymakers must consider the fact that their participation would be required not only in the development phase of DCE instrument but also in the administration phase of the final version of the instrument, which requires a larger sample size. The main consequence of this limitation is that some attributes retained in the final version of the DCE instrument may not be considered essential by policymakers; hence, they may not be statistically significant when administered in a full-scale study. Moreover, we cannot exclude the possibility that some factors that policymakers may consider important may not be adequately represented in the DCE instrument because of limitation in the variety of policymakers involved.

Third, only primigravidae were included in the interviews. The inclusion of a homogenous population of gravidae in the instrument construction was considered to limit the complexity of the experimental design. We cannot exclude the possibility that subgroups of pregnant women might value instrument attributes differently.

Finally, for practical reasons as stated in the method section, only pregnant women between 28–30 weeks' gestation were included in the testing phase of the study. These women were randomly selected from a representative sample of the population enrolled in an on-going randomized clinical trial. In addition, participants of this phase might consist of multigravida, rather than only primigravidae as in the qualitative studies. This might affect the study results.

## 5. Conclusion

Having a common DCE instrument to be administered to the beneficiaries of an intervention and policymakers is expected to allow the identification of similarities and differences in the value experiences of both groups. These quantitative data should help policymakers evaluate whether the beneficiaries' perspective that they are mandated to consider is sufficiently taken into account in their recommendations on the provision of an intervention for the population.

This study showed that the achievement of a consensus agreement from the two groups should be granted with better involvement of policymakers in each step of the DCE instrument construction. Moreover, this study demonstrated a DCE development process that allows to obtain a relative agreement on the list of attributes that could be used to construct a single DCE instrument. Therefore, a such DCE instrument can allow to compare preferences between the demand and supply sides of healthcare systems for the eventual addition of new conditions to be screened in the program. However, complete validation of the instrument is limited because there may be too few voting members of health technology assessment agencies committees to statistically ascertain the relevance of the attributes and their levels.

## Supporting information

**S1 Text. A review of the literature on Discrete-choice experiments in prenatal screening for fetal anomalies.**
(DOCX)

**S2 Text. Interview guide.**
(DOCX)

**S3 Text. DCE pilot study design.**
(DOCX)

## Acknowledgments

We would like to express our thanks to Gaétan Daigle for his valuable contribution in the data statistical analyses. We are also grateful to Sylvie Tapp and Josée Mailhot at the CHU de Québec-Université Laval for their excellent administrative support throughout the data collection.

## Author Contributions

**Conceptualization:** Hung Manh Nguyen.

**Data curation:** Hung Manh Nguyen, Carmen Lindsay, Mohammad Baradaran, Bounhome Soukkhaphone.

**Formal analysis:** Hung Manh Nguyen, Carmen Lindsay.

**Investigation:** Hung Manh Nguyen, Jason Robert Guertin.

**Methodology:** Hung Manh Nguyen, Jason Robert Guertin.

**Project administration:** Hung Manh Nguyen, Carmen Lindsay.

**Resources:** Hung Manh Nguyen.

**Software:** Hung Manh Nguyen, Mohammad Baradaran.

**Supervision:** Daniel Reinharz.

**Validation:** Jason Robert Guertin, Daniel Reinharz.

**Visualization:** Hung Manh Nguyen, Mohammad Baradaran.

**Writing – original draft:** Hung Manh Nguyen.

**Writing – review & editing:** Hung Manh Nguyen, Carmen Lindsay, Jason Robert Guertin, Leon Nshimyumukiza, Daniel Reinharz.

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
