## [Decision Letter · Decision Letter 0]

23 Feb 2023

PONE-D-22-28200Development of a discrete choice experiments questionnaire to elicit preferences by pregnant women and policymakers for the expansion of non-invasive prenatal screeningPLOS ONE

Dear Dr. Reinharz,

Thank you for submitting your manuscript to PLOS ONE. After careful consideration, we feel that it has merit but does not fully meet PLOS ONE’s publication criteria as it currently stands. Therefore, we invite you to submit a revised version of the manuscript that addresses the points raised during the review process.

Please complete the PRISMA (Preferred Reporting Items for Systematic Reviews and Meta-Analyses) checklist and include a brief report of your findings in the results section. Additionally, please complete the SRQR (Standards for Reporting Qualitative Research) checklist for the qualitative component of your study. If you choose not to include certain items from the checklists in your manuscript, please explain the reasons in your rebuttal letter.==============================

We look forward to receiving your revised manuscript.

Kind regards,

Ivan Sarmiento

Academic Editor

PLOS ONE

3.We note that the grant information you provided in the ‘Funding Information’ and ‘Financial Disclosure’ sections do not match.

“Financial support for this study was provided entirely by a grant from the PEGASUS2 project funded by Génome Canada (with Génome Québec; Génome BC; Génome Alberta; the Québec Ministère de l’Enseignement supérieur, de la recherche, de la science et de la technologie; the Fonds de recherche Québec – Santé; la Fondation de l’Université Laval; and the Centre de recherche du CHU de Québec) – Grant number: LSARP2012-4523, and the Canadian Institutes for Health Research - Grant number: GPH129342. The funders had no role in study design, data collection and analysis, decision to publish, or preparation of the manuscript. Jason R Guertin is a FRQS Research Scholar Junior 1 (Award #266460).”

Reviewers' comments:

Reviewer's Responses to Questions

**Comments to the Author**

1. Is the manuscript technically sound, and do the data support the conclusions?

Reviewer #1: Partly

Reviewer #2: Partly

2. Has the statistical analysis been performed appropriately and rigorously? 

Reviewer #1: I Don't Know

Reviewer #2: Yes

3. Have the authors made all data underlying the findings in their manuscript fully available?

Reviewer #1: Yes

Reviewer #2: Yes

4. Is the manuscript presented in an intelligible fashion and written in standard English?

Reviewer #1: Yes

Reviewer #2: Yes

5. Review Comments to the Author

Reviewer #1: 1) ABSTRACT:

a) Conclusion in the Abstract sounds more like the "outcome" or "benefit" of the study rather than the conclusion. The conclusion may be rephrased to reflect the Results of the study.

b) A very brief on the methdology of Qualitative study and the pilot study may be helpful

2) LIINE 56 - MESH terms

3) Less than 50% of the 68 women who were interested in the pilot study completed the survey. May discuss the possible reason in the Discussionsection

4) CONCUSIONS: do not reflect the results, may need re-phrasing

5) Limitations of the study might be prased clearly

Reviewer #2: This manuscript describes the development of a decision-aid to determine whether additional tests should be added to routine prenatal screening programs. The authors highlight the potentially novel development of decision-criteria that address both pregnant women and policymaker perspectives.

Overall, the manuscript is well organized and generally well-written. While the authors have done an excellent job writing the article, it would benefit from a copy-editor as some grammatical and sentence structures required me to re-read them multiple times to understand what the authors were hoping to convey to the reader (an example is lines 72-75).

All of my comments are relatively minor, but would strengthen the understanding of this work and its contributions to the field.

Conceptual comments:

1) Why were you interested in developing a consensus-based model? Understanding differences in perspectives between decision-criteria are as important as addressing similarities. It would be important to highlight the potential limitations (and benefits) of a consensus-based model and how it might shape the buy-in and use of the decision-aid.

2) It would be helpful for the authors to describe who they expect the end-users of this tool to be (e.g., whether these types of tools are usually used by policy makers, health care providers, health administrators, families)

Methods and reporting comments:

1) It would be helpful to describe why the authors chose to work only with primigravidae women. I would think people who may have already had experience with prenatal screening and being parents to children may have different opinions on testing practices. Perhaps the authors could include an overview of any literature available on this subject and/or highlight this as part of the limitations of their work.

2) The authors describe the use of a focus group to determine the selection and framing of attributes (lines 186-191) and state that this was done by email. It would be helpful to describe how a focus group methodology (where interaction between participants and collective discussion plays an important role in the method) was implemented by email. If this was more of a second consultation or modified Delphi process, readers may have an easier understanding how this step was completed.

3) It would be helpful for the authors to explain why they placed an inclusion criteria of being between 28-30 weeks gestation (whereas this was not included in development stage). (line 143)

4) It would be helpful to understand if there was any discussion of re-working the criteria around sensitivity and specificity to make these concepts more accessible to people considering the decision criteria, particularly these were priority considerations of policymakers

5) It would be helpful to describe how the cost criteria was described (e.g., was there any mention of who would cover the costs, whether it would be paid for by individual families, by public insurance or other mechanisms)

6) Lines 217: as someone who does not have expertise in creating DCE tools, it would be helpful for the authors to explain what and/or how this test tells you about how attributes are associated with choices (e.g., the logic or theory behind the analysis)

Discussion comments:

1) There seems to be a contradiction in the authors' statements on line 250, where they state that a novel element of this DEC tool is that it was reached by consensus, but they they go on to state on lie 257 that they cannot describe if and how consensus was reached on the final attributes. Perhaps this is a wording issue, but would be helpful to re-examine these statements to ensure they are consistent with one another.

While the authors did engage both pregnant women and policy makers, it is not clear that there was a rigorous process to build a consensus agreement between them. It seems the authors have prioritized the perspectives of pregnant women (as no policymakers were involved in the piloting, and relatively limited numbers involved in the development, and two of policy makers' priority considerations were excluded from the final tool). In my opinion, this manuscript offers an important first step in bridging these two perspectives, and provides an important perspective in centering patient and family perspectives. However, more needs to be done to ensure that such a tool would be recognized and trusted by the diverse types of decision-makers (policy makers and care providers) who may ultimately be the end-users of such a tool.

6. PLOS authors have the option to publish the peer review history of their article (what does this mean?). If published, this will include your full peer review and any attached files.

Reviewer #1: No

Reviewer #2: No

---

## [Author Response · Author response to Decision Letter 0]

21 Mar 2023

Editor comments: I have addressed all of journal’s requirements into my revision. Thank you.

Reviewer 1: I have incorporated all of your suggestions into my revision. They were very helpful. Thank you.

Reviewer 2: I have incorporated all of your suggestions into my revision. They were very helpful. Thank you.

---

## [Decision Letter · Decision Letter 1]

11 Apr 2023

PONE-D-22-28200R1Development of a discrete choice experiment questionnaire to elicit preferences by pregnant women and policymakers for the expansion of non-invasive prenatal screeningPLOS ONE

Dear Dr. Reinharz,

Thank you for submitting your manuscript to PLOS ONE. After careful consideration, we feel that it has merit but does not fully meet PLOS ONE’s publication criteria as it currently stands. Therefore, we invite you to submit a revised version of the manuscript that addresses the points raised during the review process.

I would insist that you provide more information regarding the results of your literature review. Specifically, I suggest that you include a brief report outlining the key findings in the results section of your paper. This is very important to situate the reader on the origin of the atributes used during the qualitative study. In addition to the brief report in the results, you can make the unpublished report available as a supplemental file. Additionally, please ensure that you have completed the appropriate checklist for reporting your literature review. 

We look forward to receiving your revised manuscript.

Kind regards,

Ivan Sarmiento

Academic Editor

PLOS ONE

Reviewers' comments:

Reviewer's Responses to Questions

**Comments to the Author**

1. If the authors have adequately addressed your comments raised in a previous round of review and you feel that this manuscript is now acceptable for publication, you may indicate that here to bypass the “Comments to the Author” section, enter your conflict of interest statement in the “Confidential to Editor” section, and submit your "Accept" recommendation.

Reviewer #2: (No Response)

2. Is the manuscript technically sound, and do the data support the conclusions?

Reviewer #2: Partly

3. Has the statistical analysis been performed appropriately and rigorously? 

Reviewer #2: Yes

4. Have the authors made all data underlying the findings in their manuscript fully available?

Reviewer #2: Yes

5. Is the manuscript presented in an intelligible fashion and written in standard English?

Reviewer #2: Yes

6. Review Comments to the Author

Reviewer #2: Many thanks for your responses and for diligently considering each of the reviewers' comments. I appreciated the opportunity to re-read your article. I do still have several comments and suggestions- again all still minor- discussed below.

Methods:

1) In the Methods section, while the literature is well-described, the authors state that they completed a systematic review. It would be helpful to know if and how they applied specific review checklists to guide this first portion of their study (e.g., did they follow PRISMA guidelines for a formal systematic reviews? scoping reviews?). Including a description of how the review complied with existing guidelines helps readers have a better understanding of the quality of the findings of the review

2) In the Discussion section, the authors state:

The main consequence of this limitation is that some attributes retained in the final version of the DCE instrument may not be considered as essential by some policymakers;, hence, they may not be statistically significant when administered in a full-scale study.

Isn't the inverse also true- that some factors that policymakers may consider important may not be adequately represented in the DCE because there was not fullsome representation (and/or variety of policymakers involved)? I would suggest stating a balanced summary of how limited policymaker participation may have affected study findings.

3) In the Discussion section, Line 144: The women that participated in the testing phase were already participants in a clinical trial on NIPS testing.

Doesn't this also likely makes them a unique group and may be worth mentioning in this same paragraph?

4) In the conclusion, Line 157, the authors state: This study showed that it is possible to obtain a consensus list of attributes from pregnant women and policymakers regarding the expansion of NIPS-based programs.

I am not convinced that the study described what you could call a consensus.

I refer back to my previous comment that while the authors did engage both pregnant women and policy makers, it is not clear that there was a rigorous process to build a consensus agreement between them. It seems the authors have prioritized the perspectives of pregnant women (as no policymakers were involved in the piloting, and relatively limited numbers involved in the development, and two of policy makers' priority considerations were excluded from the final tool). I would suggest changing this sentence to better align with what the study reported.

Grammatical comments: Overall, the article is easier to read and more clear. I also applaud the authors for writing in a language that is not necessarily their first language. However, it could still do with some improvements. I pulled out some sentences that stood out to me (and suggested some re-wording).

Within the abstract (towards the end): are you missing a word? (‘the test result presented to pregnant women is about’, this is better stated in actual results section as ‘information provided from test result', and could also be better described in table 4)

Also in abstract, the final sentence could be clearer (e.g., we did not/were not able to engage enough policymakers rather suggesting that they do not exist)

(methods) The pilot project was also used to confirm the domination choice defined from the qualitative studies that will be added to ain the full-scale study with pregnant women and policymakers, to test the plausibility of the choices made.

Perhaps better as: The pilot project confirmed the attributes to be added to the full-scale study where the plausibility of choices will be tested with pregnant women and policymakers,

(discussion) This step showed that pregnant women and policymakers have many common preoccupations besides their particular ones

Perhaps better as: This step showed that pregnant women and policymakers have both common and unique preoccupations

(discussion) This probably reflects the respect and interest of all participants for preoccupation with some attributes that were of little concern to them before the study, but whose evocation has aroused an interest

Perhaps better as: This probably reflects a changing understanding of some attributes that were of little concern before the study, but whose perceived relevance grew over the course of the study.

(line 146) In addition, these women can be whether or not primigravidae (??)

Several examples of different tenses being used in the same paragraph

7. PLOS authors have the option to publish the peer review history of their article (what does this mean?). If published, this will include your full peer review and any attached files.

Reviewer #2: No

---

## [Author Response · Author response to Decision Letter 1]

12 May 2023

We thank the reviewer for their comments and suggestions. Please find our responses to each point raised by the reviewer in uploaded file "Response to Reviewers 2023.05.10".

---

## [Decision Letter · Decision Letter 2]

12 Jun 2023

Development of a discrete choice experiment questionnaire to elicit preferences by pregnant women and policymakers for the expansion of non-invasive prenatal screening

PONE-D-22-28200R2

Dear Dr. Reinharz,

We’re pleased to inform you that your manuscript has been judged scientifically suitable for publication and will be formally accepted for publication once it meets all outstanding technical requirements.

Kind regards,

Ivan Sarmiento

Academic Editor

PLOS ONE

Additional Editor Comments (optional):

Reviewers' comments:

Reviewer's Responses to Questions

**Comments to the Author**

1. If the authors have adequately addressed your comments raised in a previous round of review and you feel that this manuscript is now acceptable for publication, you may indicate that here to bypass the “Comments to the Author” section, enter your conflict of interest statement in the “Confidential to Editor” section, and submit your "Accept" recommendation.

Reviewer #2: All comments have been addressed

2. Is the manuscript technically sound, and do the data support the conclusions?

Reviewer #2: Yes

3. Has the statistical analysis been performed appropriately and rigorously? 

Reviewer #2: I Don't Know

4. Have the authors made all data underlying the findings in their manuscript fully available?

Reviewer #2: Yes

5. Is the manuscript presented in an intelligible fashion and written in standard English?

Reviewer #2: Yes

6. Review Comments to the Author

Reviewer #2: (No Response)

7. PLOS authors have the option to publish the peer review history of their article (what does this mean?). If published, this will include your full peer review and any attached files.

Reviewer #2: No

---

## [Editor Report · Acceptance letter]

15 Jun 2023

PONE-D-22-28200R2 

Development of a discrete choice experiment questionnaire to elicit preferences by pregnant women and policymakers for the expansion of non-invasive prenatal screening 

Dear Dr. Reinharz:

I'm pleased to inform you that your manuscript has been deemed suitable for publication in PLOS ONE. Congratulations! Your manuscript is now with our production department. 

Kind regards, 

on behalf of

Dr. Ivan Sarmiento 

Academic Editor

PLOS ONE